# The lower COVID-19 related mortality and incidence rates in Eastern European countries are associated with delayed start of community circulation

Alban Ylli[1,2]*, Yan Yan Wu[3], Genc Burazeri[1,4], Catherine Pirkle[3], Tetine Sentell[3]

1 Faculty of Medicine, Department of Public Health, University of Medicine, Tirana, Albania, 2 Institute of Public Health, Tirana, Albania, 3 Office of Public Health Studies, University of Honolulu at Manoa, Honolulu, Hawaii, United States of America, 4 Department of International Health, School CAPHRI (Care and Public Health Research Institute), Maastricht University, Maastricht, The Netherlands

* albanylli@yahoo.co.uk

Data Availability Statement: Data are included in the annex as a supporting document They are available from the databases of John Hopkins Institute and European Centre of Control of

## Abstract

### Background

The purpose of this analysis was to assess the variations in COVID-19 related mortality in relation to the time differences in the commencement of virus circulation and containment measures in the European Region.

### Methods

The data for the current analysis (N = 50 countries) were retrieved from the John Hopkins University dataset on the 7th of May 2020, with countries as study units. A piecewise regression analysis was conducted with mortality and cumulative incidence rates introduced as dependent variables and time interval (days from the 22nd of January to the date when 100 first cases were reported) as the main predictor. The country average life expectancy at birth and outpatient contacts per person per year were statistically adjusted for in the regression model.

### Results

Mortality and incidence were strongly and inversely intercorrelated with days from January 22, respectively -0.83 ($p<0.001$) and -0.73 ($p<0.001$).

Adjusting for average life expectancy and outpatients contacts per person per year, between days 33 to 50 from the 22nd of the January, the average mortality rate decreased by 30.1/million per day (95% CI: 22.7, 37.6, $p<0.001$). During interval 51 to 73 days, the change in mortality was no longer statistically significant but still showed a decreasing trend. A similar relationship with time interval was found for incidence. Life expectancy and outpatients contacts per person per year were not associated with mortality rate.

Disiease https://coronavirus.jhu.edu/map.html Retreived 5 May 2020. https://www.ecdc.europa.eu/en/publications-data/download-todays-data-geographic-distribution-COVID-19-cases-worldwide Retreived 5 May 2020.

**Funding:** The authors received no specific funding for this work.

**Competing interests:** The authors have declared that no competing interests exist.

## Conclusion

Countries in Europe that had the earliest COVID-19 circulation suffered the worst consequences in terms of health outcomes, specifically mortality. The drastic social isolation measures, quickly undertaken in response to those initial outbreaks appear effective, especially in Eastern European countries, where community circulation started after March 11[th]. The study demonstrates that efforts to delay the early spread of the virus may have saved an average 30 deaths daily per one million inhabitants.

## Introduction

COVID-19 was declared a pandemic by the World Health Organization (WHO) on March 13, 2020 [1]. The WHO, in its first statement on the 22[nd] of January 2020, reported that there was evidence of human-to-human transmission of the new coronavirus identified in the Wuhan outbreak [2].

In Europe, the infection spread from China with the first cases reported in second half of January in France, Germany, Italy, Spain and the United Kingdom [3]. Sustained community circulation of SARS-Cov-2 began in late February and early March, and by the end of March, almost all European countries had reported their first 100 confirmed cases [4].

Countries in Europe started to talk about public health containment measures in late January and early February [5,6], but the majority of drastic, countrywide containment measures in Europe started in mid-March. This followed the spike of cases in Lombardy, Italy, which provided strong evidence for the devastating potential of the new virus. The WHO declared the pandemic on 11[th] of March, which gave official clarity to the scope and urgency of the issue. Containment measures varied from country to country and included such actions as closure of schools, closure of most non essential businesses and services, ban of non-essential travel, and total lockdown of cities. For most European countries, these measures had never been experienced before at such a widespread degree and intensity.

A striking difference can be seen in COVID-19 indicators between countries in Western Europe and those in Eastern Europe, with much lower cumulative incidence and mortality rates in Eastern Europe. Mortality rates range from more than 500 per million inhabitants in Spain, to less than 10 per million in Ukraine [4]. The reasons for these differences are still largely unexplained.

Recently, peer reviewed publications and other reports have explored biological factors responsible for the differences in incidence and mortality. For instance, host angiotensin-converting enzyme (ACE) receptor polymorphism and Bacillus Calmette-Guerin (BCG) vaccination have been cited as possible explanations [7,8]. While such biological factors are of important clinical significance, they are unlikely to explain the wide population differences in incidence and mortality observed across countries and regions in Europe. While there seems to be general consensus among professionals [9,10] about the overall efficacy of containment measures, an active debate remains about the effect of specific interventions such as stay at home orders and closure of all businesses [11,12].

Strong evidence about these topics is important to understand effective responses to the current pandemic and to prepare for future events.

As of the writing of this manuscript, no peer reviewed papers reported on the effect of the timing of containment measures in relation to the spread of COVID-19 across countries in Europe. Thus, a relatively straightforward analysis of the timing the epidemic spread across various European countries and its effects on mortality rates, is warranted and informative.

The study objective was to assess whether differences in COVID-19 related mortality were associated with differences in the timing of documented community circulation of the virus across European countries. We hypothesize that countries where the COVID-19 outbreak started later were in a better position to implement drastic control measures in time to minimize further spread of infection and consequent negative health outcomes in their population.

## Methods

### Data

All countries of the WHO European region with a population over 100 000 were included in the analyses (n = 50).

Our outcome variables were COVID-19 mortality and cumulative incidence on the 7th of May, 2020.

As an indicator of the initiation of SARS-Cov-2 community circulation—our primary predictor variable—we use the date of reporting the first 100 confirmed cases. The country specific COVID 19 related mortality rates and the dates countries reported their first 100 cases were retrieved from John Hopkins dataset [4]. The data were verified at the European Centre for Disease Control (ECDC) database [13].

We estimate similar epidemiological surveillance capacities among study units. Since the 27th of January, 2020, all WHO European region countries were included in a COVID-19 standardized surveillance system, coordinated by ECDC and the WHO Regional Office for Europe. By the end of January cross-border inter laboratory systems were in place to arrange testing and reporting of cases [3].

Time to the first 100 cases is a synthetic and relatively robust metric for measuring the initiation of a pandemic. This metric, the 100 first cases, has been used in relevant publications to classify early COVID-19 cases [14]. The number seems to represent a critical mass of cases documented during initial community circulation. Some publications have described transmission dynamics in samples of the first 100 COVID-19 cases demonstrating community circulation [15,16].

To compare the time differences between European countries concerning initiation of community circulation, we use the interval between the date of the country reporting its first 100 COVID-19 cases and the 22nd of January. The latter is the date when the WHO stated there was human-to-human transmission of the novel coronavirus. We call this indicator throughout the paper 'time interval' or 'days from 22nd January'. It is the main independent variable in our model.

To control for the influence of different proportions of older adults across European countries, the country average life expectancy at birth was included in the multivariate analyses. These data were retrieved from the World Bank dataset on the 7th of May 2020 [17]. To control for different health-care system utilization patterns, outpatients contacts per person per year was also included in the multivariate analyses. We used data on outpatient contacts from latest year available as retrieved from WHO dataset on the 10th of October 2020 [18]. We ran additional models using US data to see if the findings from Europe could be replicated in another context.

### Statistical analysis

Univariate descriptive statistics were used to summarize all variables. Non-parametric methods were applied for bivariate analysis. We calculated Spearman correlations to measure the strength of associations and used scatterplots with a locally weighted smoothing line to examine if there were non-linear relationships between dependent variables (i.e. mortality or

incidence), with time interval (number of days between the date when the 100 first COVID cases were reported and the 22nd of January). The scatterplots revealed a change of linear pattern at day 50 of the time interval (which corresponds to 11th of March, as the date when 100 first COVID-19 cases were reported); therefore, summary statistics were calculated for all variables for the time interval from day 31 to 50, and time interval from 51 to 73. One-way ANOVA tests were performed to calculate p-values for differences in means for all variables.

Next, we carried out bivariate and multivariable piecewise linear regression analyses [19] for mortality and incidence with independent variables 'time interval' with break point at day 50, 'life expectancy' and 'outpatients contacts per person per year'. Model diagnostics were performed to examine normality and influential data points were used to assess model performance. Model diagnostics showed that Italy, Belgium and Germany were influential countries in mortality analysis whereas Luxemburg and Iceland were influential for incidence analysis using Cook's D criteria. Residuals for both multiple regression analyses were approximately normal. Log-transformation of the two outcomes improved the normality but the adjusted $R^2$ was smaller for the mortality model. Interpretation of original measures of mortality and incidence were used for all models so that the interpretations were clear and meaningful. Finally, we conducted sensitivity analysis using data from the United States, since the total sample size is similar to the combined European countries. These analyses followed the same procedures as above and were done to assess whether the model explored for Europe could be replicated elsewhere.

## Results

Table 1 shows the summary statistics for all data and by time interval (time interval in days before or after day 50), and the Spearman correlations between variables. The mean mortality, incidence and life expectancy were higher in the time interval 31–50 compared to time interval 51–73 ($p < 0.001$). Mortality and incidence were highly intercorrelated (r = 0.84, $p < 0.001$), and positively associated with life expectancy (r = 0.75, $p < 0.001$). The correlation between mortality with time interval was -0.83 ($p < 0.001$) and -0.73 ($p < 0.001$) for incidence. Outpatients contacts per person per year had a weak association with mortality and incidence.

Fig 1 is the scatterplot of mortality and incidence per million vs. days from the 22nd of January with a locally weighted smoothing line. The figure shows that the slopes for both mortality and incidence before day 50 were steeper than during the time interval between days 51–73.

**Table 1. Summary statistics for mortality per million, incidence per million, life expectancy, and outpatients contacts per person per year for the full sample and by number of days from the 22nd of January (31–50 days or 51–73 days), and the Spearman correlation between the variables.**

| Variable (Range) | All Sample (N = 50) | Days 31–50 (N = 16) | Days 51–73 (N = 34) | p-value |
|---|---|---|---|---|
| | Mean ± SD | Mean ± SD | Mean ± SD | |
| Mortality (0.3–720) | 97.4 ± 162.6 | 234.8 ± 225.2 | 32.8 ± 52.2 | < .0001 |
| Incidence (40–6134) | 1504 ± 1526 | 2634 ± 1476 | 972.3 ± 1247.1 | 0.0001 |
| Life expectancy (71–84 years) | 78.2 ± 4.1 | 82.1 ± 1.3 | 76.3 ± 3.6 | < .0001 |
| Outpatients contacts per person per year | 6.4 ± 2.5 | 6.3 ± 2.2 | 6.4 ± 2.7 | 0.889 |
| **Spearman Correlation and p-value** | | | | |
| | Mortality | Incidence | Days from 22 Jan | Life expectancy |
| Incidence | 0.84 (p<0.001) | | | |
| Days from 22 Jan | -0.83 (p<0.001) | -0.73 (p<0.001) | | |
| Life expectancy | 0.75 (p<0.001) | 0.76 (p<0.001) | -0.81 (p<0.001) | |
| Outpatients contacts per person per year | 0.05 (p = 0.718) | 0.10 (p = 0.482) | -0.04 (p = 0.810) | -0.14 (p = 0.326) |

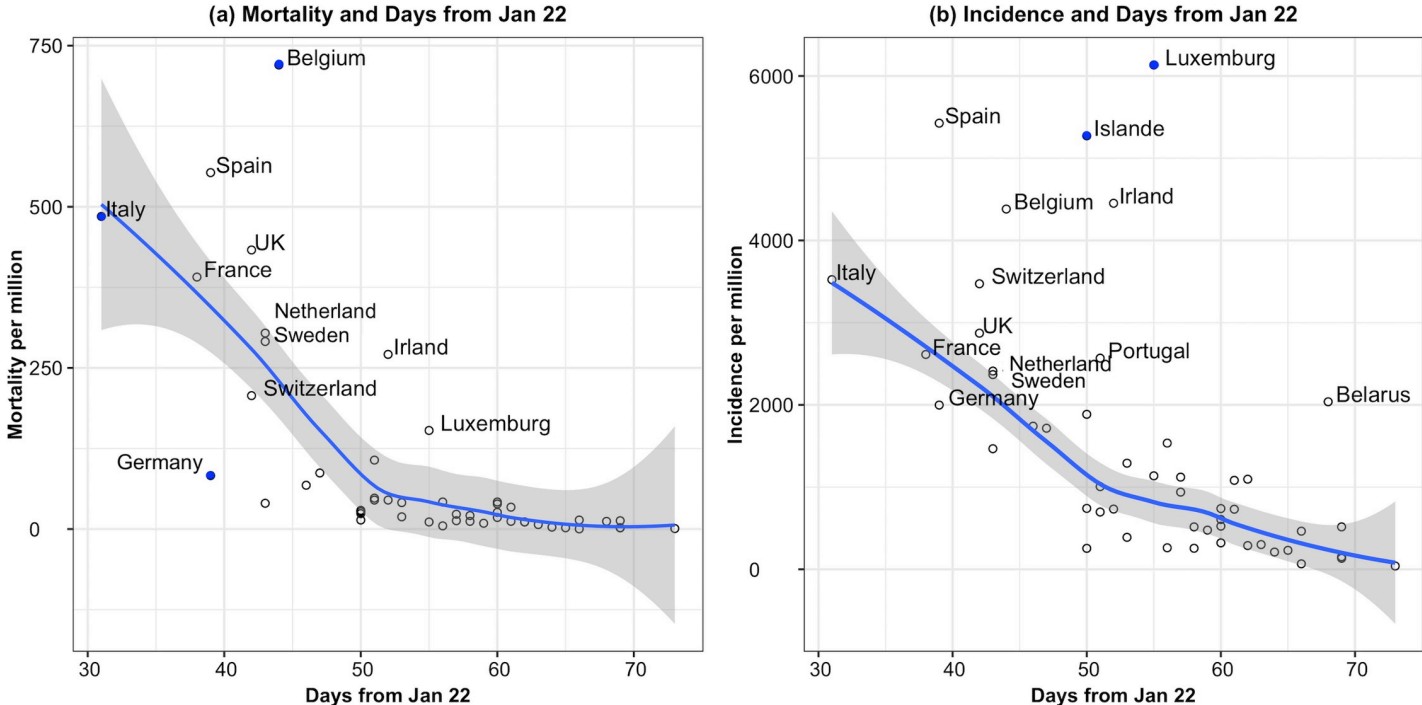

**Fig 1. Scatterplot of mortality and incidence per million versus days from the 22nd of January.** Countries labels as blue dots are influential data points using Cook's D criteria.

Results from bivariate and multivariable piecewise regression after removing influential data are displayed in Table 2. The parameter estimates for time interval attenuated slightly in the multivariable analysis, but remained statistically significant.

The multivariable analyses demonstrated that, between days 33 to 50 from the 22nd of January, the average mortality rate decreased by 30.1/million per day (95% CI: 22.7, 37.6, $p<0.001$). During interval 51 to 73 days, the change in mortality was no longer statistically significant, but still showed a decreasing trend (Beta = -0.05, 95% CI: -5.60, 5.50, p = 0.985). Life expectancy and outpatient contacts per person per year were not associated with mortality rate. A similar relationship with time interval was found for incidence; however, in contrast to the

**Table 2. Bivariate and multivariable regression analysis of mortality and incidence per million with three independent variables: Days from the 22nd of January with break point at day 50 (time interval 31–50 and 51–73 days), life expectancy, and outpatients contacts per person per year.**

|  | Bivariate Analysis | | Multivariable Analysis | |
|---|---|---|---|---|
|  | **Beta (95% CI)** | ***p*-value** | **Beta (95% CI)** | ***p*-value** |
| *Mortality per million* |  |  | *Adjusted $R^2$ = 71%* |  |
| Time interval (31–50 days) | -31.24 (-38.14, -24.34) | <0.001 | -30.14 (-37.64, -22.65) | <0.001 |
| Time interval (51–73 days) | -2.33 (-5.70, 1.05) | 0.171 | -0.05 (-5.60, 5.50) | 0.985 |
| Life expectancy (in years) | 17.17 (9.58, 24.76) | <0.001 | 5.26 (-4.31, 14.83) | 0.282 |
| Outpatients contacts per person per year | -3.35 (-18.48, 11.77) | 0.664 | 2.67 (-5.74, 11.08) | 0.534 |
| *Incidence per million* |  |  | *Adjusted $R^2$ = 56%* |  |
| Time interval (31–50 days) | -153.23 (-221.58, -84.88) | <0.001 | -122.67 (-194.61, -50.74) | <0.001 |
| Time interval (51–73 days) | -62.52 (-106.76, -18.29) | 0.007 | -2.04 (-71.77,67.69) | 0.953 |
| Life expectancy (in years) | 213.88 (146.44, 281.31) | <0.001 | 147.29 (22.92, 271.66) | 0.020 |
| Outpatients contacts per person per year | -0.33 (-19.68, 19.02) | 0.973 | 53.07 (-49.45, 155.60) | 0.310 |

**Table 3. Regression analysis for USA states: Mortality per million with independent variable days from the 22nd of January with break point at day 59 (time interval 46–59 and 60–70 days).**

| (a) Mortality per million in the U.S. on May 7th | | |
|---|---|---|
| | Beta (95% CI) | p-value |
| Time interval (46–59 days) | -35.40 (-56.87, -13.93) | 0.001 |
| Time interval (60–70 days) | -12.10 (-34.55, 10.36) | 0.284 |
| (b) Mortality per million in the U.S. on Oct 3 | | |
| | Beta (95% CI) | p-value |
| Time interval (46–59 days) | -35.61 (-67.03, -4.20) | 0.026 |
| Time interval (60–70 days) | -39.21 (-72.06, -6.36) | 0.020 |

mortality model, higher life expectancy was found to be associated with higher incidence (Beta = 147.3, 95% CI: 22.9, 271.7, $p$ = 0.02). The adjusted $R^2$ (proportion of variation explained by the model) was 75% for mortality and 60% for incidence.

Table 3 and Fig 2 show the results from the sensitivity analyses using data from the United States. These analyses demonstrate very similar findings to those reported from Europe.

## Discussion

Timing matters. Results from this study indicate that the time when sustained virus circulation started in a country is associated with the health impact of the pandemic for that country. Overall, later sustained circulation in a country was associated with lower overall mortality and incidence rates.

Results from these analyses of 50 European countries also reveal that the relationship between mortality and incidence rates and time of commencement of community circulation of SARS-Cov-2 does not follow a linear gradient. Mortality and incidence rates decreased

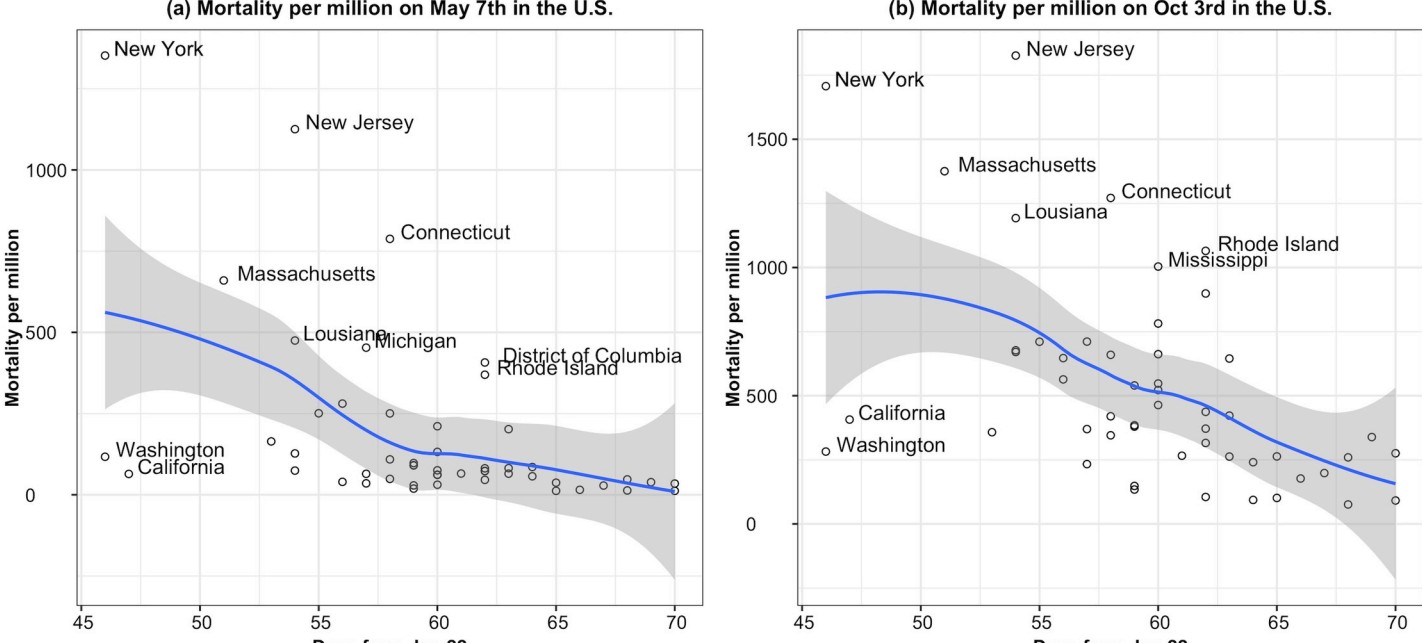

**Fig 2. Scatterplot for USA data: Mortality per million versus days from the 22nd of January.**

significantly for every day of delay until the 50[th] day from the 22[nd] of January (i.e. March 11[th]). After that point, the gradient became less steep and time becomes non significant.

The non linear nature of the relationship may reflect the fact that in the days after the declaration of world pandemic on March 13, 2020, countries were quick to introduce drastic containment measures. The majority of countries, where pandemic started after mid-March, likely benefited similarly from the timely measures to contain viral spread and the differences between them became insignificant. Google mobility reports confirm the quick effects on social distancing of the mid-March measures [20].

The countries in Europe that observed the earliest COVID-19 circulation, suffered the worst consequences in terms of health outcomes, specifically mortality. While only a matter of a few weeks difference, this time interval may have changed dramatically the reservoir of virus circulating in the community and thereby the trajectory of incident disease and resulting deaths.

Reaching the first 100 cases in a country depends also on seeding episodes. Large urban areas of Western Europe have more intensive global connection, more international travel, and a higher potential for more seeding events, which increased the chances of earlier community transmission in these countries [21].

The longer that it took for a country to get to 100 reported cases, the easier it may have been to control the epidemic using public health policy action. The drastic social isolation measures undertaken in European countries where community circulation of the virus started after March 11[th] seem to have been well-timed. This may explain their significantly lower COVID-19-related incidence and mortality in Eastern European countries compared with the Western European countries.

On March 10[th] and 11[th], the Italian government was the first in Europe to issue decrees [22,23] introducing countrywide lockdown measures, limiting movement out of the home and banning the operation of a number of businesses (bars, retail shops etc). On March 20[th] the measures were further tightened, banning all non essential open air walking. During a short period, after the 11[th] of March, most European countries introduced similar measures [24]. Evidence suggests that these measures were effectively enforced and that community mobility was significantly reduced [20]. As the disease spread, drastic measures in Eastern European countries, and in the Western 'periphery' of Europe (i.e.Portugal), appear to have been effective in mitigating COVID-19 mortality. In the West, where community circulation had initiated much earlier, the measures taken in mid-March (or even later) were comparatively late and allowed a large mass of COVID-19 cases, building a critical reservoir of infection in the population. Consequently, the efficacy of public health actions was greatly reduced and the most vulnerable members (older adults, those with chronic conditions) of society were deeply affected.

While some governments in Asia (i.e. China) had already taken drastic public health measures to effectively curb the epidemic [25], we assume that after the declaration of a global pandemic,decision-makers in Europe were in a better position to take and enforce such extreme measures, which only weeks before had seemed too draconian. International mass media coverage of the pandemic outcomes on the Italian health system and the high risk of dying in Italian northern regions, also influenced quick decision-making by political leadership by mid-March. In countries with swift responses, including those fortunate to have experienced later community spread and fewer seeding events, the outbreaks were less pervasive and the most vulnerable less affected.

As the pandemic is ongoing, there may be small observed changes in the health outcome differences documented here, with regard to the timing of a critical mass of cases in various regions of Europe. However, it is highly unlikely they will significantly change in the

associations documented during the first wave of pandemic. With few exceptions (for example Russia), as of May 7th, in European countries the epidemic curves were flattened, the epidemic peaks were past, and the effective reproductive numbers were around 1 [26].

We also tested the validity of the proposed model with 50 states of United States of America using the same source of data. Our results showed a similar pattern and very similar statistics for the two interval gradients. Using the U.S. data, the cut-off was only about one week later compared to Europe (59 days from 22nd January, or 19th of March). We also applied the model using the 3d of October mortality data and the direction of the trend remained the same; although, the association becomes more linear. The later findings most probably confirm the weaken of the early effect of spring measures.

While life expectancy is lower in the Eastern Europe [17] and a potential confounder in analyses comparing different European countries with varying life expectancies, the multivariate analyses show that timing of the outbreak was a more important factor for mortality. Further, the burden of chronic disease is higher in Eastern compared to Western Europe [27], potentially putting Eastern European populations at higher risk of mortality from COVID-19. Country level results indicate, however, that Eastern European countries have fared better than their Western European counterparts indicating that relative chronic disease burden is an improbable explanatory factor for differences in country level outcomes.

A number of other factors can be discussed to explain differences in COVID-19 outbreak trajectories between Western and Eastern Europe, including urbanization and population density. Currently, these factors are controversial [28,29] and no publications demonstrate consistent evidence to support these as deciding factors.

Compulsory BCG vaccination programs across countries have also been associated with COVID-19 morbidity and mortality in a geographical correlation study [8], but the WHO states that there is insufficient evidence to confirm this [30]. A trial testing the potential effect of BCG vaccines to boost immunity against COVID-19, is underway in Germany and the Netherlands [31]. Geographical variations of ACE2 receptor polymorphism has also been reported as a possible explanation to COVID-19 epidemiological findings [7]. Nonetheless, the two last hypotheses cannot explain the important intra country variations observed in Italy or elsewhere. The delay in infection and timeliness of the measures may better explain the much lower COVID-19 morbidity and mortality rate in Sicily, Sardinia or other regions in 'periphery' of Italy, compared to the Northern regions where first clusters where reported.

In conclusion, the results of this study provide secondary evidence about the effect of public health measures taken in Europe after the 11th of March, when pandemic was declared by WHO. They demonstrate that efforts to delay the early spread of the virus might save daily an average 30 deaths per one million inhabitants. The study can help public health professionals to better understand the pattern of pandemic spread and its relation to pandemic mortality. The results will support decision makers during pandemic to take early and swift public health measures in order to assure important health benefits in subsequent months.

Our study is subject to several limitations. Our model may explain some important country differences; for example, the slope in the first regression segment is apparently driven by lower mortality in countries with community circulation reported during 8-11th March (from Austria to Czech Republic). It does not explain all observed differences, such as that between Belgium and Germany or Sweden and Norway. Further research, focused on comparing specific country situations, is needed in the future.

We examine mortality and cumulative incidence as they are reported by countries. The data may have issues which can be only partially validated. While incidence is highly affected by country testing strategies, the reported mortality has been used as a valid health outcome in other studies [7,8] We also use the date of first 100 cases, as they are reported by countries

health authorities. Similar sources of data have been considered valid to systematically document community COVID-19 cases [14]. We know that silent community circulation of the virus started before initial detection. Nonetheless, the observation of country epidemic curves seems to generally confirm the ranking of reported initiation dates [32]. Further, genetic sequencing methodology has been used to track early outbreaks in Europe and the USA and it supports the timeline pattern provided by early documented cases [33].

We acknowledge that the ecological study method used in the analyses has limitations and does not allow conclusions on causality. We include only a few variables, and two outcomes. We recommend that future analyses include more potential covariates to explore the complex causal web of COVID-19 health outcomes and their relationships with policy decisions and provide estimates of effects of specific factors on outcomes. Finally, while our hypothesis about higher efficacy to control the epidemic where it started later seems logical, more research is needed on specific public health measures taken by European countries. Such research should find ways how to standardise the interventions and how to make use of relevant non-English literature.

## Supporting information

**S1 Data. Annex countries data.**
(DOC)

## Author Contributions

**Conceptualization:** Alban Ylli, Catherine Pirkle, Tetine Sentell.

**Data curation:** Alban Ylli, Yan Yan Wu, Genc Burazeri.

**Formal analysis:** Alban Ylli, Yan Yan Wu, Genc Burazeri.

**Methodology:** Alban Ylli, Yan Yan Wu, Catherine Pirkle.

**Supervision:** Tetine Sentell.

**Validation:** Catherine Pirkle.

**Writing – original draft:** Alban Ylli, Genc Burazeri.

**Writing – review & editing:** Alban Ylli, Yan Yan Wu, Genc Burazeri, Catherine Pirkle, Tetine Sentell.

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
