## [Decision Letter · Decision Letter 0]

25 Sep 2020

PONE-D-20-15377

The lower COVID-19 related mortality and incidence rates in Eastern European countries are associated with delayed start of community circulation

PLOS ONE

Dear Dr. Ylli,

Thank you for submitting your manuscript to PLOS ONE. After careful consideration, we feel that it has merit but does not fully meet PLOS ONE’s publication criteria as it currently stands. Therefore, we invite you to submit a revised version of the manuscript that addresses the points raised during the review process.

Your manuscript was reviewed by 2 experts in the field. Both identified significant problems in your submission. Please review the attached comments and provide point-by-point responses.

We look forward to receiving your revised manuscript.

Kind regards,

Yury E Khudyakov, PhD

Academic Editor

PLOS ONE

Journal Requirements:

Reviewers' comments:

Reviewer's Responses to Questions

**Comments to the Author**

1. Is the manuscript technically sound, and do the data support the conclusions?

Reviewer #1: Partly

Reviewer #2: No

2. Has the statistical analysis been performed appropriately and rigorously? 

Reviewer #1: Yes

Reviewer #2: No

3. Have the authors made all data underlying the findings in their manuscript fully available?

Reviewer #1: Yes

Reviewer #2: No

4. Is the manuscript presented in an intelligible fashion and written in standard English?

Reviewer #1: No

Reviewer #2: Yes

5. Review Comments to the Author

Reviewer #1: Review PLOS One: The lower COVID-19 related mortality and incidence rates in Eastern European countries are associated with delayed start of community circulation

Thank you for the opportunity to review this interesting manuscript.

In general, please proofread this manuscript carefully. There are multiple small typos, missing or extra words, and missing or extra punctuation. Given there are no line numbers, it is unwieldly to note them all in the specific comments below.

Abstract – background

This sentence is long and unwieldly and should be simplified or separate into two sentences.

Abstract – conclusion first line

No comma after “circulation”

Introduction first paragraph, second sentence

“statement of on” – delete “of”

Introduction first paragraph, second sentence

Structure of the sentence makes it unclear whether human-to-human transmission or whether the novel coronavirus was first described to the WHO on Dec 31.

Introduction second paragraph, second sentence

Citations needed

Introduction third paragraph, second sentence

Should read something like “These measures varied from country to country and included such actions as” – otherwise, it implies all actions were taken in all places, which is not true.

Introduction third paragraph, last sentence

“size” is an incorrect word here – perhaps “widespread degree”

Page three, final paragraph

Should read “For instance, host angiotensin…” – these are not the only biological differences noted and should not be implied as such.

Page four, first paragraph, final sentence

You mention “measures” – to what are you referring?

Page four, second paragraph, final sentence

This sentence is not grammatical nor is it clear.

Page five, second paragraph

This sentence is convoluted. Please simplify or separate into two sentences.

Page five, fourth paragraph

March 11th was the first 100 cases where?

Page six, first paragraph

Is the interval 51-71 or 51-73? It is mentioned as both in the previous paragraphs. Why was 71/73 chosen as a cut off? It appears that all countries reported 100 cases by this time period, if so, please specify in the body of the manuscript.

Page seven, final sentence

Was higher or lower life expectancy associated with incidence?

Discussion

Please carefully proofread – there is a lot of non-standard English here.

You need citations throughout the discussion, including for “seeding events”, “intensive global connection”.

Your paragraph on life expectancy and chronic disease in Western v Eastern Europe is difficult to understand. Please clarify.

Page ten, final paragraph

Should be “intra country variations observed in Italy” not “inner”

General comments

How do you control for different rates and implementation dates of testing in these fifty countries? You do address incidence/mortality but you do not address how implementation dates of testing may wildly vary your date of first 100 confirmed cases per country. This may significantly impact the outcome of your analyses.

How do you control for different health-care system utilization across these countries?

How do you control for racial differences across these countries?

How do you control for socio-economic differences?

Why do you think there was a statistically significant difference in days 31-50 and not days 51-73?

Given that the virus can have an incubation period of up to 14 days post exposure, was there a significant change in viral transmission 14 days after individual countries imposed their containment measures? Was there a change in trend of time to 100 cases in the 14 days after Mar 11?

Reviewer #2: The manuscript examines the variations in COVID-19 mortality and incidence rates in relation to the delayed start of community virus circulation in Eastern European countries. While there are some potentials in this study, I have concerns about the novelty, social significance, and policy implications of this study, and therefore I would suggest resubmission. Authors need to address the following comments:

--The modeling without validation is not reliable. The authors need to examine whether their model is consistent with independent (test) dataset/recent data or not. With such a small sample size, the associations are not unlikely and not very reliable.

--Page 5 last line: I don’t think model diagnostics are used to improve “model fit”? This is used mostly to check the underlying assumptions. Also, I don’t see any accuracy assessments done for the model.

--The authors need to justify based on what criterion they have selected the first 100 cases as the main predictor. I mean, why not 90,80, 10? It may cause significant differences in the results. What is the scientific reason behind it?

--Country-level age structure, racial diversity, and socio-economy should be considered as confounders.

-What is the significance and broad impacts if the gap is filled? What is the added value of this work to public health? How the results can help public health decision-makers in practice.

6. PLOS authors have the option to publish the peer review history of their article (what does this mean?). If published, this will include your full peer review and any attached files.

Reviewer #1: No

Reviewer #2: No

---

## [Author Response · Author response to Decision Letter 0]

19 Oct 2020

Answers to Reviewer #1 

In general, please proofread this manuscript carefully. There are multiple small typos, missing or extra words, and missing or extra punctuation. Given there are no line numbers, it is unwieldly to note them all in the specific comments below.

RESPONSE: We proofread all the manuscript carefully, corrected all errors and improved the English as needed. 

Abstract – background

This sentence is long and unwieldly and should be simplified or separate into two sentences.

RESPONSE: Done. We simplified it.

Abstract – conclusion first line

No comma after “circulation”

RESPONSE: Corrected as suggested 

Introduction first paragraph, second sentence

“statement of on” – delete “of”

RESPONSE: Corrected as suggested 

Introduction first paragraph, second sentence

Structure of the sentence makes it unclear whether human-to-human transmission or whether the novel coronavirus was first described to the WHO on Dec 31.

RESPONSE: We revised the sentence and clarified it is about human-to-human transmission. 

Introduction second paragraph, second sentence

Citations needed

RESPONSE: Reference provided

Introduction third paragraph, second sentence

Should read something like “These measures varied from country to country and included such actions as” – otherwise, it implies all actions were taken in all places, which is not true.

RESPONSE: True. We corrected it following the reviewer suggestion. 

Introduction third paragraph, last sentence

“size” is an incorrect word here – perhaps “widespread degree”

RESPONSE: Corrected as suggested.

Page three, final paragraph

Should read “For instance, host angiotensin…” – these are not the only biological differences noted and should not be implied as such.

RESPONSE: Corrected as suggested.

Page four, first paragraph, final sentence

You mention “measures” – to what are you referring?

RESPONSE: We specified the measures under debate (stay at home orders and closure of all businesses)

Page four, second paragraph, final sentence

This sentence is not grammatical nor is it clear.

RESPONSE: We corrected it.

Page five, second paragraph

This sentence is convoluted. Please simplify or separate into two sentences.

RESPONSE: We separated the concept into two sentences. 

Page five, fourth paragraph

March 11th was the first 100 cases where?

RESPONSE: This describes the distribution of time interval (days from the date the first 100 cases were documented from January 22). The change of the linear pattern was found to be at interval time 50. This interval time corresponds to March 11th. 

It is the main independent indicator in our model and its full name is a bit too long and complex. It creates some difficulties in sentences. In the revised manuscript we clarify the terminology. We also standardize its use throughout the text.

Page six, first paragraph

Is the interval 51-71 or 51-73? It is mentioned as both in the previous paragraphs. Why was 71/73 chosen as a cut off? It appears that all countries reported 100 cases by this time period, if so, please specify in the body of the manuscript.

RESPONSE: 71, in previous paragraph was an error. We corrected it to 73. Yes, the range of the time interval is 51-73 days. 

Page seven, final sentence

Was higher or lower life expectancy associated with incidence?

RESPONSE: Higher life expectancy. We specified it in the revised text. 

Discussion

Please carefully proofread – there is a lot of non-standard English here.

RESPONSE: We carefully proofread the English.

You need citations throughout the discussion, including for “seeding events”, “intensive global connection”.

RESPONSE: Now we provide more citations, including for seeding and global connection.

Your paragraph on life expectancy and chronic disease in Western v Eastern Europe is difficult to understand. Please clarify.

RESPONSE: Corrected 

Page ten, final paragraph

Should be “intra country variations observed in Italy” not “inner”

RESPONSE: Corrected as suggested

General comments

How do you control for different rates and implementation dates of testing in these fifty countries? You do address incidence/mortality but you do not address how implementation dates of testing may wildly vary your date of first 100 confirmed cases per country. This may significantly impact the outcome of your analyses.

RESPONSE: This is an important point. Thank you. Since the 27 of January, all WHO European region countries were included in a COVID-19 standardized surveillance, coordinated by the European Centre for Disease Prevention and Control (ECDC) and the WHO Regional Office for Europe.By the end of January, cross-border inter laboratory systems were in place to arrange testing and reporting of cases. (#3 in the references). We now describe this on page 5, second paragraph.

We also know that middle-income countries outside the European region, such as Iran, where the pandemic seems to have started earlier than in most Europe, had demonstrated capacities to identify COVID-19 cases since early in February. Thus, the ability to detect COVID-19 in Europe existed by the end of January.

In the discussion we also mention the observation of country’s epidemiological COVID-19 curves, which confirm the trend of pandemic initiation timeline in specific countries, as expressed by 100 first official cases. Curbs show that outbreaks in France, Spain and Germany are slightly delayed compared to Italy, while the pandemic in UK starts even later. Additionally, while Russia had already confirmed its first 2 COVID-19 cases at the end of January, the community circulation there (first 100 cases), started only in March. The ‘Russian’ curb reached the peak much later than most countries in west. 

A recent relevant publication based on viral genetic sequence data backs the main timeline pattern provided by early confirmed cases in Europe and USA. We cite this now in the discussion on page 14, second paragraph. 

(https://science.sciencemag.org/content/early/2020/09/11/science.abc8169)

Despite its limitations, time of 100 first cases seem to be a synthetic and relatively robust indicator for measuring the initiation of pandemic. Other recent relevant publications are using it to indicate early cases. 

(https://www.thelancet.com/journals/laninf/article/PIIS1473-3099(20)30581-8/fulltext). 

In the revised version we elaborate the issue and the metric more extensively in the methods and discussion. We also cite more relevant references 

How do you control for different health-care system utilization across these countries?

RESPONSE: We agree this is an important point. To address this question, we looked at outpatient contact per person per year in the European Region. Upon observation, this variable does not appear to be a meaningful confounder. For example, countries of Ex Soviet Union have the highest per capita utilization of health care, and Nordic countries the lowest. Also, in Belarus, Hungary or Lithuania this indicator is much higher than in most European Union. Yet, utilization rates do not map onto COVID-19 mortality or incidence rates. 

Nevertheless, because utilization patters could affect study findings, in the revised version of the paper, we have included this new variable in the model. Neither the pattern nor the magnitude of association between time interval and mortality and incidence rates change, when including this variable.

How do you control for racial differences across these countries?

RESPONSE: In the United States in particular, mortality rates have been higher for certain racial groups, including Blacks, Hispanics and Pacific Islanders. Higher mortality for these groups likely reflects poorer underlying health, lower access to health services and poorer quality of care when health services are accessed. This pattern may also be the case for Europe. However, for race to be a confounder it must be associated with BOTH the independent and dependent variables. While there is certainly evidence of an association between race and mortality, such evidence is lacking for the independent variable:time interval (days from the 22nd of January to the date when 100 first cases were reported). It is hard to imagine how the racial make-up of a country would affect the time between the 22nd of January and the date when the first 100 cases were reported. As such, it does not appear appropriate to control for a country’s racial make-up. 

Further, we ran additional models using US data to see if the findings from Europe could be replicated in another context. They were replicated, despite a very different racial/ethnic make-up in the US compared to Europe (more details below). 

How do you control for socio-economic differences?

RESPONSE: We estimate that the most important component of socio-economic differences in COVID-19 mortality is health care utilization, instead of indicators such as World Bank GDP per capita. As mentioned above we looked carefully at healthcare utilization patterns and included the new variable in the new model. Similar to the response above, we also ran additional models using US data to see if the findings from Europe could be replicated in another context. They were replicated, despite notable socioeconomic differences between European countries and states in the USA (more details below). 

Why do you think there was a statistically significant difference in days 31-50 and not days 51-73?

RESPONSE: We think the non linear nature of the relationship is produced by the fact that in the days after the declaration of world pandemic (corresponding to 11thof March or 51st day in our model), countries were quick to introduce the drastic containment measures. The majority of countries, where pandemic started after mid March, benefitted similarly from the timely measures and the differences among them became insignificant. Google mobility reports confirm quick effects on social distancing of mid March measures. We have now indicated this on discussion, page 10, third paragraph.

We also tested the validity of the model with 50 states of United States of America. It shows a similar pattern (and very similar statistics for the two intervals), with the cut off only about one week later. We also applied the model using the 3rd of October mortality and the direction of the trend remained the same, although the association becomes linear (most probably confirming the weakening of the early effect of spring measures). Here below are the results of our model for USA 50states.

(a) Mortality per million in the U.S. on May 7th 

 Beta (95% CI) p-value

Time interval (46-59 days) -35.40 (-56.87, -13.93) 0.001

Time interval (60-70 days) -12.10 (-34.55, 10.36) 0.284

(b) Mortality per million in the U.S. on Oct 3 

 Beta (95% CI) p-value

Time interval (46-59 days) -35.61 (-67.03, -4.20) 0.026

Time interval (60-70 days) -39.21 (-72.06, -6.36) 0.020

We mention these extra analyses in the methods, results, and discussion. 

Given that the virus can have an incubation period of up to 14 days post exposure, was there a significant change in viral transmission 14 days after individual countries imposed their containment measures? Was there a change in trend of time to 100 cases in the 14 days after Mar 11?

RESPONSE: We didn’t look at it in this analysis. The ecologic study methodology we have applied has its limitations. We acknowledge it in the revised paper. 

Answers to Reviewer #2: 

The modeling without validation is not reliable. The authors need to examine whether their model is consistent with independent (test) dataset/recent data or not. With such a small sample size, the associations are not unlikely and not very reliable.

RESPONSE: Thank you. This is an important point. We have limited the analysis to the dataset of European Region Countries for following reasons:

It encompasses a substantial number of countries (16) where early COVID-19 cases start circulating before 11th of March, when pandemic is officially declared.

There was a standard surveillance system put in place by the WHO

During the revision, we tested the validity of the model with 50 states of United States of America (where only 3 states had documented 100 cases before 11th of March). It shows a similar pattern (and very similar statistics for the two intervals), with the cut off only about one week later compared to Europe. We also applied the model using the 3d of October mortality and the direction of the trend remained the same, although the association becomes linear (most probably confirming the weakening of the early effect of spring measures). Here below are the results of our model for USA.

(a) Mortality per million in the U.S. on May 7th 

 Beta (95% CI) p-value

Time interval (46-59 days) -35.40 (-56.87, -13.93) 0.001

Time interval (60-70 days) -12.10 (-34.55, 10.36) 0.284

(b) Mortality per million in the U.S. on Oct 3 

 Beta (95% CI) p-value

Time interval (46-59 days) -35.61 (-67.03, -4.20) 0.026

Time interval (60-70 days) -39.21 (-72.06, -6.36) 0.020

We mention these extra analyses in the methods, results, and discussion. 

We also encourage other researchers to apply it at country level, using regions as study unit. 

We agree that the ecological study method is an issue, as it is the sample size, and have acknowledged it at limitations of revised paper on page 14, third paragraph. 

We also agree that datasets based on official information published online, may have issues which can be only partially validated. Nonetheless, they remain an important source of information and are being used in a number of relevant publications. 

(https://www.thelancet.com/journals/laninf/article/PIIS1473-3099(20)30581-8/fulltext)

We cite more relevant references in the revised manuscript.

--Page 5 last line: I don’t think model diagnostics are used to improve “model fit”? This is used mostly to check the underlying assumptions. Also, I don’t see any accuracy assessments done for the model.

RESPONSE: We elaborated more on model diagnostics and model fitting. 

Model diagnostics were performed to examine normality and influential data points and adjusted R2 statistics were used to assess model performance. Model diagnostics showed that Italy, Belgium and Germany were influential for mortality analysis whereas Luxemburg and Island were influential for incidence analysis using Cook’s D criteria. Residuals for both multiple regression analysis were approximately Normal. Log-transformation of the two outcomes improved the Normality but adjusted R2 were smaller for mortality model. Interpretation of original measures of mortality and incidence were used for all models so that the interpretations were more meaningful. 

We reflected this in the statistical methods section and the title of the figure. 

The authors need to justify based on what criterion they have selected the first 100 cases as the main predictor. I mean, why not 90,80, 10? It may cause significant differences in the results. What is the scientific reason behind it?

RESPONSE: This is an important point. Thank You. In our model, this metric is crucial to quantify early or delayed start of COVID-19 community circulation. 

As pandemic is underway and more research is carried out, the 100 first cases are used in relevant publications to classify early COVID-19 cases.

(https://www.thelancet.com/journals/laninf/article/PIIS1473-3099(20)30581-8/fulltext). 

The indicator seems to represent a critical mass of cases documented during initial community circulation. A number of publications have described transmission dynamics in samples of first 100 COVID-19 cases demonstrating community circulation. 

(https://www.ncbi.nlm.nih.gov/pmc/articles/PMC7195694/)

(https://www.cdc.gov/mmwr/volumes/69/wr/mm6911e1.htm)

Still, it may seem arbitrary and its use is yet to be validated. We acknowledge this at revised paper limitations.

We also cite more relevant references in the revised manuscript.

Country-level age structure, racial diversity, and socio-economy should be considered as confounders.

RESPONSE: We agree this is an important point. To address this question, we looked at outpatient contact per person per year in the European Region. Upon observation, this variable does not appear to be a meaningful confounder. For example, countries of Ex Soviet Union have the highest per capita utilization of health care, and Nordic countries the lowest. Also, in Belarus, Hungary or Lithuania this indicator is much higher than in most European Union. Yet, utilization rates do not map onto COVID-19 mortality or incidence rates. 

Nevertheless, because utilization patters could affect study findings, in the revised version of the paper, we have included this new variable in the model. Neither the patter nor the magnitude of association between time interval and mortality and incidence rates, change when including this variable.

Now, in the revised version of the paper we have included the new variable in the model. The pattern and the power of association don’t change. 

Race: In the United States in particular, mortality rates have been higher for certain racial groups, including Blacks, Hispanics and Pacific Islanders. Higher mortality for these groups likely reflects poorer underlying health, lower access to health services and poorer quality of care when health services are accessed. This pattern may also be the case for Europe. However, for race to be a confounder it must be associated with BOTH the independent and dependent variables. While there is certainly evidence of an association between race and mortality, such evidence is lacking for the independent variable: time interval (days from the 22nd of January to the date when 100 first cases were reported). It is hard to imagine how the racial make-up of a country would affect the time between the 22nd of January and the date when the first 100 cases were reported. As such, it does not appear appropriate to control for a country’s racial make-up. 

We proxy age structure with life-expectancy at birth.

What is the significance and broad impacts if the gap is filled? What is the added value of this work to public health? How the results can help public health decision-makers in practice.

RESPONSE: The study provides secondary evidence about the effect of public health measures taken in Europe after 11th of March, when pandemic was declared by WHO. It demonstrates that efforts to delay the early spread of the virus, may save daily an average 30 deaths per one million inhabitants. 

We think the non linear nature of the relationship between early start and mortality is produced by the fact that in the days after the declaration of world pandemic (corresponding to 11th of March or 51st day in our model), countries were quick to introduce the drastic containment measures. The majority of countries, where pandemic started after mid March, benefit similarly from the timely measures and the differences among them become insignificant. 

The results help public health professionals to better understand the pattern of pandemic spread and its relation to pandemic mortality. It also provides systematic evidence against speculations about pandemic reaching its peak since January in countries of East Europe, what explained according to them, the very low mortality in late April.

The resultswill support decision makers during pandemic to take early and swift public health measures in order to assure important health benefits in subsequent months.

We elaborated more extensively this issue in the discussion section of the revised version.

---

## [Decision Letter · Decision Letter 1]

23 Nov 2020

The lower COVID-19 related mortality and incidence rates in Eastern European countries are associated with delayed start of community circulation

PONE-D-20-15377R1

Dear Dr. Ylli,

We’re pleased to inform you that your manuscript has been judged scientifically suitable for publication and will be formally accepted for publication once it meets all outstanding technical requirements.

Kind regards,

Yury E Khudyakov, PhD

Academic Editor

PLOS ONE

Additional Editor Comments (optional):

Reviewers' comments:

Reviewer's Responses to Questions

**Comments to the Author**

1. If the authors have adequately addressed your comments raised in a previous round of review and you feel that this manuscript is now acceptable for publication, you may indicate that here to bypass the “Comments to the Author” section, enter your conflict of interest statement in the “Confidential to Editor” section, and submit your "Accept" recommendation.

Reviewer #2: All comments have been addressed

2. Is the manuscript technically sound, and do the data support the conclusions?

Reviewer #2: Yes

3. Has the statistical analysis been performed appropriately and rigorously? 

Reviewer #2: Yes

4. Have the authors made all data underlying the findings in their manuscript fully available?

Reviewer #2: (No Response)

5. Is the manuscript presented in an intelligible fashion and written in standard English?

Reviewer #2: (No Response)

6. Review Comments to the Author

Reviewer #2: Please use the space provided to explain your answers to the questions above. You may also include additional comments for the author, including concerns about dual publication, research ethics, or publication ethics. (Please upload your review as an attachment if it exceeds 20,000 characters) (Limit 100 to 20000 Characters)

"The authors addressed my comments"

7. PLOS authors have the option to publish the peer review history of their article (what does this mean?). If published, this will include your full peer review and any attached files.

Reviewer #2: No

---

## [Editor Report · Acceptance letter]

25 Nov 2020

PONE-D-20-15377R1 

The lower COVID-19 related mortality and incidence rates in Eastern European countries are associated with delayed start of community circulation 

Dear Dr. Ylli:

I'm pleased to inform you that your manuscript has been deemed suitable for publication in PLOS ONE. Congratulations! Your manuscript is now with our production department. 

Kind regards, 

on behalf of

Dr. Yury E Khudyakov 

Academic Editor

PLOS ONE